# Vaccines Against COVID-19: A Review

**DOI:** 10.3390/vaccines10030414

**Published:** 2022-03-10

**Authors:** Carlos U. Torres-Estrella, María del Rocío Reyes-Montes, Esperanza Duarte-Escalante, Mónica Sierra Martínez, María Guadalupe Frías-De-León, Gustavo Acosta-Altamirano

**Affiliations:** 1Hospital Regional de Alta Especialidad de Ixtapaluca, Ciudad de México PC 56530, Mexico; carlos.torres0118@gmail.com (C.U.T.-E.); sierrammtz@gmail.com (M.S.M.); magpefrias@gmail.com (M.G.F.-D.-L.); 2Unidad Profesional Interdisciplinaria de Biotecnología, Instituto Politécnico Nacional (IPN), Ciudad de México PC 07340, Mexico; 3Departamento de Microbiología y Parasitología, Facultad de Medicina, Universidad Nacional Autónoma de México (UNAM), Ciudad de México PC 04510, Mexico; remoa@unam.mx (M.d.R.R.-M.); dupe@unam.mx (E.D.-E.); 4Escuela Superior de Medicina, Instituto Politécnico Nacional (IPN), Ciudad de México PC 11340, Mexico

**Keywords:** SARS-CoV-2, COVID-19, vaccines, immunity, efficacy, humoral immunity, cellular immunity

## Abstract

As a result of the COVID-19 pandemic, various joint efforts have been made to support the creation of vaccines. Different projects have been under development, of which some are in the clinical evaluation stage and others in are in phase III with positive results. The aim of this paper was to describe the current situation of the development and production of vaccines available to the population to facilitate future research and continue developing and proposing ideas for the benefit of the population. So, we carried out a systematic review using databases such as PubMed, ScienceDirect, SciELO, and MEDLINE, including keywords such as “vaccines,” “COVID-19,” and “SARS-CoV-2”. We reviewed the development and production of the anti-COVID vaccine and its different platforms, the background leading to the massive development of these substances, and the most basic immune aspects for a better understanding of their physiological activity and the immune response in those who receive the vaccine. We also analyzed immunization effects in populations with any medical or physiological conditions (such as immunosuppression, people with comorbidities, and pregnancy), as well as the response to immunization with heterologous vaccines and the hybrid immunity (the combination of natural immunity to SARS-CoV-2 with immunity generated by the vaccine). Likewise, we address the current situation in Mexico and its role in managing the vaccination process against SARS-CoV-2 at the national and international levels. There are still many clinical and molecular aspects to be described, such as the duration of active immunity and the development of immunological memory, to mention some of the most important ones. However, due to the short time since the global vaccination roll-out and that it has been progressive (not counting children and people with medical conditions), it is premature to say whether a second vaccination schedule will be necessary for the near future. Thus, it is essential to continue with health measures.

## 1. Introduction

Until recently, the coronavirus family was not considered highly contagious. Human coronaviruses, such as HCoV-229E, HCoV-OC43, HCoV-NL63, and HCoV-HKU1, are very common and usually cause colds like other pathogens but have low immunogenicity [1]. However, since SARS-CoV appeared in 2002–2003 and MERS-CoV in 2012, health authorities worldwide began to pay attention to the behavior and dispersal of these genotypes. As a result, various research lines emerged to understand better their pathogenicity and immunogenic capacity [2].

At the end of 2019, a new disease was announced caused by a viral agent belonging to the coronavirus family. It was named SARS-CoV-2 due to its relationship with the virus that causes Severe Acute Respiratory Syndrome (SARS) and the infection mechanism is relies on the protein “S”, which is composed of two subunits, S1, which contains the receptor-binding domains that recognizes and binds to the host receptor angiotensin-converting enzyme 2 (ACE-2), and subunit S2, which mediates viral cell membrane fusion [3]. Its infection causes the disease COVID-19, an acronym for coronavirus disease with the year it was discovered. The first registered case occurred in Wuhan, China, on 31 December 2019 [4]. Shortly afterward, the World Health Organization (WHO) declared a pandemic due to the rapid spread of this new virus worldwide, affecting thousands of people. As a result of this pandemic, various joint efforts have been made to support the creation of vaccines. At least 300 projects have been reported under development, of which 40 are in clinical evaluation and 10 in phase III with positive results. In addition, to prevent the spread of the virus and protect the population, the WHO has licensed at least five vaccines for emergency use [5].

## 2. Methodology

A literature review about vaccines against COVID-19 was carried out. The databases used in the search were Scopus, PubMed, ScienceDirect, MEDLINE, and SciELO. The search was performed on each database using a combination of the following keywords: “vaccines,” “SARS-CoV-2,” and “COVID-19”. As inclusion criteria, only references with full text in English or Spanish were considered. The titles and abstracts of the references were reviewed as exclusion criteria to assess their coherence with the search topic, and unrelated articles were dismissed.

## 3. COVID-19 Vaccines Development

Developing a vaccine is a complex process that can take up to 8 years before being available to the population. In addition, strict protocols must be followed to guarantee its safety and efficacy. However, for COVID-19, it was completely different. There is a 79.5% genetic similarity between SARS-CoV and SARS-CoV-2 [6]. Thus, the previous knowledge gathered from the first investigations on this virus family that began almost two decades ago proved to be relevant for developing vaccines against SARS-CoV-2 and, of course, producing them in record time. Additionally, the contributions of Katalin Karikó regarding the use of modified mRNA as a tool for immunological purposes [7] allowed the global pharmaceutical industry to start producing these vaccines. A brief scheme of the entire vaccine development process and COVID-19 vaccine is shown in Figure 1.

The development of the different types of vaccines shows the immense technological advances that exist today (Table 1). However, like any tool, they have advantages and disadvantages (Table 2) that must be pondered before choosing the most appropriate one. Moreover, vaccines can also enhance the immune response with the help of substances called adjuvants.
vaccines-10-00414-t001_Table 1Table 1Platforms used for the development and production of vaccines.Type of VaccineDevelopmentInvention YearTarget DeseaseAttenuated pathogenThrough physicochemical treatments, the pathogen loses features that allow an effective infection. Due to the intact antigens on the membrane surface, they can be recognized by the immune system.1798SmallpoxDead/inactivated pathogenThrough physicochemical treatments, the bacterial pathogen is killed and viral pathogen is inactivated. It cannot infect, but the antigens must remain on the membrane to be recognized by the immune system.1896 TyphoidToxoidsThe bacterial toxins are attenuated with chemical agents such as formaldehyde or the effects of heat, preserving their high immunogenicity.1923 DiphtheriaProtein subunitsThey contain only harmless proteins of the microorganism. They are made by recombinant expression in cell models such as bacteria or fungi, or obtained by lysis of the pathogen, but the proteins that join to the host’s receptors are preserved to protect their three-dimensional conformation.1970 AnthraxViral particlesThe structural proteins of the pathogen are assembled using a matrix (which can be a lipid bilayer) which allows simulating the pathogen’s spatial conformation without genetic content. They have high immunogenicity since most of the pathogen’s proteins are present.1986 Hepatitis BViral VectorsThese are genetically modified viruses, which have already been well characterized. The genetic content is eliminated, except for those genes that give a cell the ability to infect. The removed genetic material is replaced by that that is of interest (DNA or mRNA *) and incorporated into the virus for protection and transport. Once the vector comes into contact with human cells, it instructs them to produce a protein exclusive for the microorganism. Thus, the body begins to manufacture components of the immune system. Most of these viral vectors cannot replicate.2019 EbolaNucleic acidsThey can be DNA or mRNA. In both cases, the genetic material is protected by a nanoparticle, mainly lipids, since it becomes permeable to the phospholipid bilayer of the cell membrane. DNA travels through the cytosol until incorporated into the nucleus, where it is transcribed into mRNA and later translated into a chain of amino acids. Something similar happens with the mRNA, but it does not enter the nucleus, instead passsing directly to the ribosomes to synthesize the chain of amino acids. Finally, this genetic material allows the production of pathogenic proteins which will be expressed at the membrane surface level, thus achieving the creation of antigens through our cells, which will stimulate the immune system.2020 SARS-CoV-2Refs. [8,9]; * DNA: Deoxyribonucleic acid; mRNA: Messenger-type ribonucleic acid.
vaccines-10-00414-t002_Table 2Table 2Advantages and disadvantages of different platforms for vaccine development.PlatformAdvantagesDisadvantagesAttenuated pathogenProduces humoral and cellular response with a single dose.Safety problems in immunosuppressed people.Strains are difficult to obtain.Dead/inactivatedpathogenSafe due to the nature of its composition.Very easy to transport and store.Large amounts of the pathogen.Possible effects on the immunogenicity of the antigenProtein subunitsSafe during production and for immunosuppressed people.Decrease in APC * capacity due to particle size.Limited production due to product scalability.PolysaccharidesAlternative against bacterias with abundant polysaccharide antigens.There is only IgM production.Low memory immunity.Low efficiency in children.Viral particlesCombines the efficacy of live and subunit vaccines.High scalability production.Particle assembly is a complex process.Viral VectorsIt can induce a humoral and cellular response.Safe.Pre-existing immunity is used against the vector.It needs low temperatures to store.Nucleic acidScalability.Rapid design and development.Very secure. Induces humoral and cellular responses.Its storage and handling are delicate.Refs. [5,10]; * APC: Antigen presenting cell.
Figure 1(**A**) Sequential diagram of the vaccine development process. The exploratory stage is the basis for the evolution of the product under development; finding an antigen that correctly stimulates the immune system allows the rest of the process to continue. Human testing is the most time-consuming stage due to the number of individuals involved in the study. In addition, a wide range of regions must be included to assess environmental and genetic conditions and detect if there are differences between study groups. (**B**) COVID-19 vaccine development is one of the greatest accomplishments in medical history. Due to the vast contribution of scientific research in SARS-CoV, the production of this vaccine was faster compared to others. The time described is according to the clinical trials for the different vaccines. Source: [11,12,13] Design: Carlos U. Torres-Estrella.
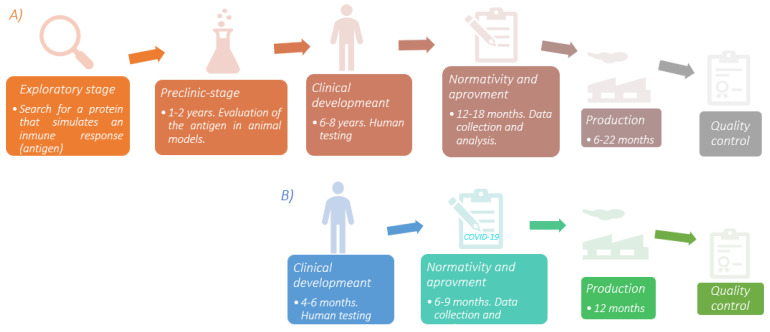



### Adjuvants

Aluminum salts as adjuvants have been used in different vaccines, such as the PiCoVacc and CoV-RBD219N1 vaccines for the SARS-CoV-2 virus, promoting higher antibody production [13]. The Novavax vaccine uses Matrix-M1 as an adjuvant, based on purified saponin obtained from the Quillaja saponaria tree, formulated with cholesterol nanoparticles and phospholipids [14]. This adjuvant enhances the immune response by recruiting a larger amount of APC at the application site, which increases the presentation of antigens to T cells, generating a TH1 and TH2 response [15]. In addition, it induces antibodies of multiple subclasses, mainly high titers of anti-S IgG that block the binding of the ACE2 receptor, neutralizing the virus, reducing the dose of antigen used in each vaccine, and reducing costs [16]. SARS-CoV-2 vaccines formulated with soluble homogeneous “Spike” protein S trimers (S-2P) use AS03 as an adjuvant, an oil-in-water emulsion composed of squalene, polysorbate 80, and α-tocopherol. This compound induces an IgG antibody response sufficient to protect from exposure to SARS-CoV-2 [17]. The Sclamp vaccine, which is in the preclinical stage, consists of stabilized recombinant viral “Spike” glycoprotein S in its trimeric form used in combination with the MF59 adjuvant, which consists of a 4.3% squalene oil emulsion in water, stabilized with Tween 80 and Span 85. This adjuvant interacts with the APCs at the inoculation site and participates in the transfer of antigens, increasing the efficiency of antigen presentation and stimulating a significant immune response mediated by neutralizing antibodies [18]. The CIGB2020 compound, produced by the Center for Genetic Engineering and Biotechnology (CIGB) in Havana, Cuba, has been used to formulate vaccines against hepatitis. It stimulates specific genes to produce interferons and other cytokines with antiviral properties, which also intervene in the expression of class I and II human leukocyte antigen (HLA) molecules. These molecules are involved in the activation of the innate immune system and increase the presentation of viral antigens, and therefore, stimulate T cells [19]. In March 2020, clinical trials were initiated to explore and evaluate the immunotherapeutic effect of this substance in suspected and confirmed patients with SARS-CoV-2 infection. However, the results of this protocol have not been published yet [20].

## 4. COVID-19 Vaccines

The available information about this coronavirus family was immensely advantageous compared to other cases since, as explained in Figure 1, the time required to formulate a vaccine is considerable. Table 3 and Table 4 list the main vaccines currently available and those used since the beginning of 2021 in the global vaccination campaign to counteract the number of infections and victims of COVID-19 disease.

Most of these vaccines have been approved by the WHO with an “emergency” use license; that is, these are vaccines that present conclusive data regarding the level of conferred protection, safety, and efficacy when monitoring the population selected for the clinical trials for at least two months [37]. However, on 23 August 2021, and 31 January 2022, the FDA approved with a full license the use of Pfizer/BioNTech and MODERNA vaccines, respectively [38,39]. It also must be considered that public and private investments, which are unprecedented events, allowed the great researchers of the 21st century to act immediately to develop these vaccines. Unfortunately, some of these products have encountered problems during application in various regions. The first complication occurred with the Pfizer/BioNTech vaccine. A case was reported of a person who developed an exacerbated allergy to one of the primary vaccine compounds, polyethylene glycol, an allergenic molecule for certain people. Therefore, it is recommended to conduct a comprehensive review of the patients’ medical history to detect any known allergies [22]. On the other hand, after receiving the Astra Zeneca/Oxford vaccine, the presence of blood clots that caused either thrombosis or thrombocytopenia was reported in certain patients. This incident led some countries to suspend vaccination with this vaccine, even though 20 million doses had already been applied. Shortly after, on 19 March 2021, the WHO made a public statement in conjunction with the Pharmacovigilance Risk Assessment Committee (PRAC) of the European Medicines Agency (EMA) regarding these events. The statement read that all adverse events had been carefully analyzed and that pondering the number of vaccine doses already applied, the number of reported cases was not relevant to conclude that the vaccine was unsafe. Additionally, the number of reports was within the permissible limits for these very rare events. Finally, they stressed that the benefits provided by the vaccine were more significant than the possible side effects that may arise and that it is essential to assess the clinical conditions of each patient for these events [40].

Still, many other projects are in the early stages despite financing issues and specialized human resources needs. These enterprises will undoubtedly promise a very competitive market, in which purchasing power will not be a problem for less developed countries as there will be an extensive manufacturer catalog [41].

### 4.1. Dose Immunization

Viral vector and nucleic acid vaccines require the application of two doses. The reason why this vaccination scheme is needed is that the immune system works in two ways. The first time there is contact with a pathogen, nonspecific IgM-type antibodies are produced (Figure 2A) that will help to block the pathogen during the primary response, which begins within three to five days. Then, highly specific and low-molecular-weight IgG-type antibodies (Figure 2B) are produced 10 to 15 days later. This adaptive response can last weeks and even months [42] (Figure 3). Because it is a parenteral type of immunity, the production of these immunoglobulins in the bloodstream is stimulated to avoid possible damage at the systemic level [43], along with a small synthesis of IgA immunoglobulins.

The genetic material contained in the vaccine is limited to the number of copies supplied to the person receiving it. Not all will translate into a protein, as some will degrade due to vesicular trafficking. Thus, a second dose ensures that the adaptive response is activated more efficiently and that the number of translated proteins is enough for a response. However, this is not strong enough to keep immune levels up. Due to the fact that the immune response decreases, a third dose is necessary to maintain the protection (Figure 3) according to the Center for Disease Control and Prevention (CDC) and the Johns Hopkins medical school [44,45].
Figure 3Production of IgM and IgG in serum, and IgA in mucosal surface antibodies from the first and second doses, simulating a first exposure to the pathogen and its immune memory. The stimulation of antibody IgA has been demonstrated after 1st dose vaccination [46]. Third dose guarantees to remain protected after six months. Design by: Carlos U. Torres-Estrella.
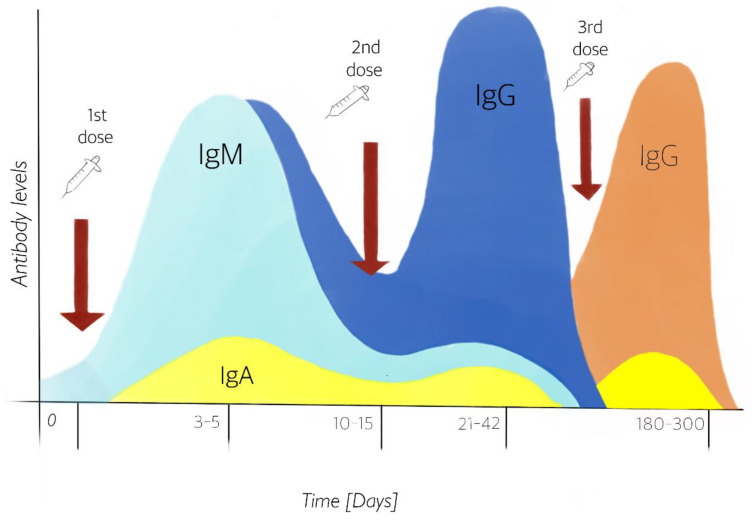



### 4.2. Heterologous Vaccines

It is known that booster immunization with heterologous vaccines can increase the intensity and amplitude of immune responses. Therefore, in the face of the current pandemic caused by the SARS-CoV-2 virus, heterologous vaccines against COVID-19 have emerged as a valuable alternative to provide the necessary immunological protection to the affected population. He et al. [47] conducted a study aimed at maximizing the benefits of vaccination using a heterologous booster strategy in a murine model, applying different combinations of four types of leading candidate vaccines against COVID-19 that were found in clinical trials in China. Their results showed that sequential immunization with an adenovirus-vectored vaccine followed by administering an inactivated/recombinant subunit/mRNA vaccine specifically increased neutralizing antibody levels and promoted the modulation of immune response antibodies to predominantly neutralizing antibodies. In addition, the heterologous priming and boosting regimen with an adenovirus vector-based vaccine also enhanced Th1-biased T cell responses. Thus, their results provided new ideas for developing and applying COVID-19 vaccines to control the SARS-CoV-2 pandemic.

Likewise, a study conducted by Hillus et al. [48], evaluated the reactogenicity and immunogenicity of heterologous immunizations with ChAdOx1 nCov-19 (AstraZeneca, Cambridge, UK) and BNT162b2 (Pfizer-BioNTech, Mainz, Germany), in comparison with homologous immunization with BNT162b2 and ChAdOx1 nCov-19. The study showed evidence of the safety and immunogenicity of heterologous ChAdOx1 nCov-19-BNT162b2 vaccination, currently recommended in several countries. Thus, their data support ongoing efforts to investigate heterologous vaccination regimens for COVID-19.

On the other hand, Eroglu et al. [49] mentioned that the preliminary results of a trial with CombiVacS that included more than 600 patients in Spain showed the benefits of mixing different vaccines against coronavirus. In addition, they mentioned that the combination of the Oxford–AstraZeneca vaccine and the Pfizer–BioNTech vaccine also induced a stronger immune response than two doses of the same vaccine. However, they emphasized that it is essential to consider that some safety concerns remain when two different vaccines are combined as each vaccine has its own set of adverse events/side effects. Nevertheless, no severe side effects were reported in the combined trials.

## 5. Humoral and Cellular Immunity Generated by Vaccines

As already mentioned, the objective of vaccines is to achieve an effective and risk-free immunization without clinical complications, except for the common side effects (fever, pain, inflammation in the injection site, and headache). However, rare side effects are reported and investigated immediately [50]. Table 5 briefly shows the immunological characteristics acquired after SARS-CoV-2 vaccination. Although there are unreported data due to the clinical stage of some projects, it is observed that the majority fulfill the task of generating immunity and conferring protection to patients. Regarding mucosal-associated immunity, it is known that parenteral immunization can induce a response on this surface. Additionally, some vaccines stimulate the migration of T and B cells to these compartments [51], although this is not the case for SARS-CoV-2 vaccines. However, a substantial proportion of IgG antibodies against protein S and its RBD (Receptor Binding Domain) and IgA antibodies were reported in saliva samples of 45 healthcare workers after receiving the second dose of the Pfizer and Moderna vaccines [52]. It was also described that the reactivity of IgA and IgG antibodies in saliva and serum, as well as IgM, increased rapidly after vaccination in two people who had already contracted COVID-19. However, studies have not yet been conducted to quantify the production of immunoglobulins in the mucosa nor its duration.

### 5.1. Pregnant Women and Vaccination against COVID-19

A report was published recently on a small number of pregnant women (35,691 participants) between the ages of 16 and 54 who received the COVID-19 mRNA vaccine. The women experienced many typical side effects of a vaccine immunization, such as headaches, myalgias, chills, and fever. However, none presented any clinical alteration of greater risk. Pain at the injection site was also reported more frequently in pregnant women than in those who are not pregnant [53]. However, more extensive clinical trials to support these facts are not ruled out. In addition, it must be kept in mind that each woman evolves differently during pregnancy, so it is essential to evaluate the patient’s current health condition upon vaccination.

### 5.2. Population with Medical Conditions and COVID-19 Vaccination

The COVID-19 vaccine can be administered to people under medical treatment that does not contraindicate to vaccination. Patients with comorbidities can proceed with immunization [54], and receive any of the above vaccines. However, people with a medical condition such as immunosuppression or an autoimmune disease must consider certain factors, although there are still no sufficient clinical data on this population on their evolution after receiving the COVID-19 vaccine. According to CDC, immunosuppressed people can receive the COVID-19 vaccine since the clinical trials included HIV patients with immunosuppressive conditions. Although data from this population are not yet sufficient, they are supported by the “general practices for the vaccination of the immunocompromised population”, which also apply to people receiving immunosuppressive therapy such as chemotherapy. The only condition is that the vaccination schedule must be completed at least two weeks before starting treatment. Individuals with advanced conditions can receive the vaccine if they undergo a medical assessment that includes the type of vaccine, immunosuppression levels, the time it will last, and any comorbidities [55]. On the other hand, patients undergoing dialysis have shown greater immunogenicity with the MODERNA vaccine, up to 12 weeks after second dose vaccination [56].

### 5.3. Hybrid Immunity

In June 2020, it was revealed that the SARS-CoV-2 virus began to generate mutations, allowing it to evade neutralizing antibodies and become more infectious. Some of the mutant viruses that completely replaced the original SARS-CoV-2 were B.1.1.7 (Alpha), B.1.351 (Beta), B.1.1.248 (Gamma), and B.1.617.2 (Delta), which were called variants of interest (VOC) [57]. Among these, the Delta variant is the one that is spreading throughout the world. The presence of mutations in most VOC results in partial escape from antibodies. This biological process is evidence of selection pressure to evade natural immunity.

A reduced neutralization potential of the antibodies against variants has been observed in the ChAdOx1 nCoV-19 vaccine (AstraZeneca, Cambridge, UK) against symptomatic patients. It was observed that the efficacy decreased from 75% to 11% against variant B.1.351 [58]. The BNT162b2 vaccine (Pfizer-BioNTech, Mainz, Germany) efficacy against symptomatic cases of the B.1.351 variant fell from ~95% to 75%, and protection in severely ill patients remained at 97% [59]. Likewise, it has been observed that both vaccines retain most of their efficacy against the B.1.617.2 (Delta) variant. Due to the reduction in neutralizing antibodies, the B cells recognize the mutated spike proteins of the variants. Studies of natural infection with B.1.351 have shown that the B cells recognize the mutated spike proteins of this variant since the responses of neutralizing antibodies were robust against that variant and the ancestral strain [60]. Therefore, it is crucial to understand the immune response mechanisms to vaccination.

On the other hand, several researchers have shown that the vaccination of people convalescing from COVID-19 can produce neutralizing antibodies that can be up to a thousand times higher than those induced by infection or vaccination, suggesting that a way to control the pandemic may be the induction of hybrid immunity [61,62,63,64,65]. This type of immunity results from a combination of natural immunity and immunity generated by the vaccine. A greater than expected immune response arises when innate immunity to SARS-CoV-2 is combined with vaccine-induced immunity. “Hybrid immunity” is particularly interesting because people with prior SARS-CoV-2 infection develop unexpectedly strong immune responses to vaccines [61,66]. The latter is because memory B cells have two functions: one is to produce clonal antibodies after reinfection with the same virus, and the other is to encode a library of antibody mutations to protect themselves from viral variants that may emerge in the future. Thus, a substantial proportion of memory B cells encode antibodies capable of binding or neutralizing VOC, and the quality of those memory B cells increases over time [64,65].

Additionally, Andreano et al. [67] analyzed the repertoire of B cells that produce neutralizing antibodies after vaccination with BNT162b2 mRNA in healthy individuals not previously infected with SARS-CoV-2 and people with prior exposure to SARS-CoV-2. Subjects with previous SARS-CoV-2 infection responded to vaccination with a higher number of antibody-producing B cells not susceptible to variants of interest and with higher neutralizing power. The latter may be partly explained by the increased number of somatic mutations and the fact that seropositive individuals shed antibodies potent germline-derived IGHV2-5 and IGHJ4-1 antibodies, which were not observed in healthy vaccinated subjects [65]. Thus, it is suggested that the third dose of this vaccine could lead to a response similar to hybrid immunity since neutralizing antibodies after infection and vaccination are mainly derived from the same immunodominant germ lines [68].

## 6. Immunization Alternatives

Routes of mucosal immunization, such as intranasal and oral, are constantly being studied. However, their application in clinical practice has been limited due to the technical and safety challenges they pose. Most currently approved human vaccines are administered systemically, e.g., intramuscular and subcutaneous. Although these pathways can elicit systemic humoral immune responses and are regulated by antigen-specific cells, they are generally incapable of producing IgA responses and mucosal protective immunity. The mucosal pathway of immunization causes immune responses at the local and distal mucosal sites and systemic immune responses. Therefore, most current efforts to obtain mucosal protective immunity have focused on mucosal vaccination routes, such as oral and intranasal [69]. Although the induction of a mucosal response by systemic immunization remains poorly understood, in recent studies, the existence of neutralizing antibodies against SARS-CoV-2 in saliva, the pharynx, and the nose has been shown using swabs to collect the samples [70]. Therefore, using an appropriate adjuvant could change the outcome and lead to IgA expression [71]. Currently, intranasal vaccines against SARS-CoV-2 are being developed (Figure 4) but are still in the preclinical study stage. However, it has been observed that they can generate certain immunity in the mucosa since there is a production of IgA antibodies, which bind SARS-CoV-2 RBD with high affinity, involving interaction with the ACE2 receptor [72].

## 7. Mexico and Vaccination against COVID-19

In Mexico, the lethality rate for COVID-19 in 2021 was 8.7, which places the country in first place worldwide on this indicator [73]. However, it is expected that these figures will decrease as the vaccination campaign progresses. So far, 80 million doses have been administered, with only 70 million people fully vaccinated as of January 2022, equaling 60.62% of the total population [74].

Given that the current demand for vaccines is extremely high, some pharmaceutical companies, such as Pfizer-BioNTech (Mainz, Germany), have not been able to supply all the orders requested by the various countries, including Mexico. Therefore, they have found themselves needing to open new manufacturing sites to achieve a successful supply [75]. Likewise, Mexico has ventured into performing some parts of the manufacturing process locally to counteract the shortage. Such is the case for the Astra Zeneca [76] and CanSino Biologics vaccines, in which the packaging is performed locally to benefit the population from gaining access more quickly [77,78].

On the other hand, Mexico has decided to develop its vaccine. Four projects are known to be under development (Table 6), which were presented to the Coalition for Innovations in Preparedness for Epidemics (CEPI). Although they are still undergoing preclinical evaluation, it is expected that they will reach phase III with promising results. This enterprise will allow the country to enter a very competitive market niche and demonstrate its great scientific capacity.

Of all projects, the one with the greatest potential is the one that Avimex^®^ will produce. This vaccine will use the technology developed by the Icahn School of Medicine at Mount Siani in New York City and the University of North Carolina at Chapel Hill. Their experiments focused on the expression of protein S in chicken embryos using the Newcastle virus as a viral vector [79]. This laid the foundations to scale the production of this vaccine and partner with Avimex^®^. This company has over a decade of experience producing drugs for veterinary use and even a vaccine against the A(H1N1) pandemic influenza virus [80].

## 8. Discussion

Vaccination against SARS-CoV-2 has been an important tool to protect the population, including children 12–15 years old, from severe illness and long-term complications However, vaccination in children under 12 years old has been controversial because, although there are multiple benefits, there are also potential risks [81]. Recently, a clinical study evaluated two mRNA vaccines against SARS-CoV-2 in children younger than 12 years of age showing the safety, immunogenicity, and efficacy of vaccination in this pediatric population [82]. So, in a short time, the entire pediatric population may be vaccinated.

The introduction of vaccines is just a step forward to ending the current pandemic. We must not fail to follow the prevention measures established by the CDC, which include wearing protective glasses and masks and constant handwashing, among others. Additionally, it is vital to implement new strategies that reduce the exposure and contagion of the virus, such as nasal masks [83]. Along with the vaccination campaign, these measures will help counteract the virus’s spread until there is a sufficient availability of all vaccines. Likewise, it is necessary to obtain more data on the cellular immunity these vaccines induce and how rapidly they produce immunological memory since most data point to humoral immunity. The current data lead us to believe that, although current vaccines are being developed quickly, they confer a low level of immunogenicity, which is why a poor duration of cellular immunity has been seen.

On the other hand, the recently described variants pose a challenge for researchers as they may be more infectious [84], and very few vaccines have reported effectiveness against these mutations. Due to the above, vaccination schedules must be carefully analyzed to determine if a long-lasting and protective immunity is created or if booster doses, a combination of immunogens, or even annual vaccinations will be required. It also raises the question of whether these vaccines will be sufficient to counteract the spread of the virus and its new strains in the long term or if present platforms will need to be adjusted to provide an adequate immunogenic level. The latter does not mean that current platforms fail to fulfill their mission fully. However, being the first time in the history of medicine that these types of vaccines are used, it is necessary to make modifications to improve active immunity and guarantee that it is present in the majority of the population, including children and people with medical conditions. In this context, one component that can be added to vaccines to induce an immune response is adjuvants. The most important are aluminum salts, formulated with protein S, inducing neutralizing antibodies and protecting the body against SARS-CoV-2. However, various studies have shown that it cannot activate CD4 + T and CD8 + T cell responses.

On the other hand, the adjuvants based on emulsions (MF59, AS03), and the Toll-like receptor (TRL) agonists, unlike aluminum salts adjuvants, can provoke a more balanced immune response in protein-based vaccines. The previous is because they improve the antigen uptake and recruit immune cells, inducing both a humoral and a cellular response, which could be more beneficial [85,86]. Although various adjuvants have been used to formulate COVID-19 vaccines, more studies are required to achieve a combination of antigens and adjuvants that can provide a safer and more effective result. Likewise, another alternative to reduce exacerbated inflammation in patients could be the immunopotentiation of vaccines, or therapeutic vaccines, given when the individual is already infected. For instance, these vaccines are used for Human Papilloma Virus (HPV) to stimulate a better cellular response, mainly CD4 + and CD8 + lymphocytes [87], due to these immune cells’ natural low activation. Something similar happens with SARS-CoV-2 infection, raising the possibility of using these types of vaccines to promote a better response in the body.

On the other hand, an important issue is epigenetics. Even though no studies have delved into this issue yet, we believe that evaluating it is essential because of the variability in vaccine efficacy in different regions. Another factor to consider is the nutritional status of the individual. The main aim of vaccines is to achieve an effective, long-lasting, and protective immune response in the body, depending on various cells and molecules such as lymphocytes, antibodies, and cytokines. These cells require adequate energy sources, substrates, and nutrients, derived mainly from diet for their synthesis, development, and correct functioning [88].

The immune system needs energy-producing substrates to activate. Glucose and amino acids are required to generate cellular structures, fatty acids to produce lipid derivatives such as prostaglandins and leukotrienes, and vitamins and minerals serve as cofactors with an immunomodulatory effect. All these components are acquired through diet, which, if inadequate, cause an ineffective immune response to natural infection or induced by a vaccine [89].

Malnutrition favors an inadequate immune system response due to the lack of nutrients and substrates for the synthesis of immunological components. For instance, vitamin A intervenes in the maturation of T lymphocytes and the proliferation of CD8 +lymphocytes and B lymphocytes. Malnutrition can have two extremes that affect the immune response, malnutrition, and obesity. Obesity is an abnormal or excessive accumulation of adipose tissue, considered an endocrine organ, leading to a state of immunosuppression. Such a state is associated with the production of adipokines, mainly leptin, which under physiological conditions participate in the activation of neutrophils, proliferation of T cells, and cytokine synthesis. In obese individuals, high leptin levels induce a state of resistance that reduces physiological activities. Other immunological alterations related to obesity are decreased circulating lymphocyte populations and the hyperactivation of the mTOR protein in immune cells, which favors an increased activation of effector cells, compromising memory lymphocytes production [90]. Therefore, this population needs to improve its eating habits since vulnerable individuals have some nutritional deficits, as seen throughout this pandemic.

However, we are aware of the immense effort that has been made over just one year. We acknowledge that this generational leap in vaccine development represents one of the most significant technological advances in medicine of the last decade. This breakthrough will support new immunization routes that better activate the immune response. We consider that one of these routes should be the mucous membranes, as it is the first route of contact with many pathogens, and it is necessary to neutralize them and avoid compromising the lower respiratory tract or decreasing its damage. Likewise, the United States government has requested to release the patent for vaccines against COVID-19 [91], which could enhance its development and upgrade. Additionally, pharmaceutical companies using the same platform will be able to conduct more detailed studies that allow the combination of vaccines, increasing the number of individuals with complete vaccination schedules in a shorter time. The previous also implies a boost in their production. Another option is to support projects that improve the immunogenicity of available vaccines and reduce the number of doses, achieve broader coverage, and above all, provide developing countries the opportunity for rapid access. In the meantime, we must not ignore the fact that detailed knowledge on these new technologies is still limited to many people, so it is necessary to take preventive measures that allow safe production and high-quality levels and standards.

## Figures and Tables

**Figure 2 vaccines-10-00414-f002:**
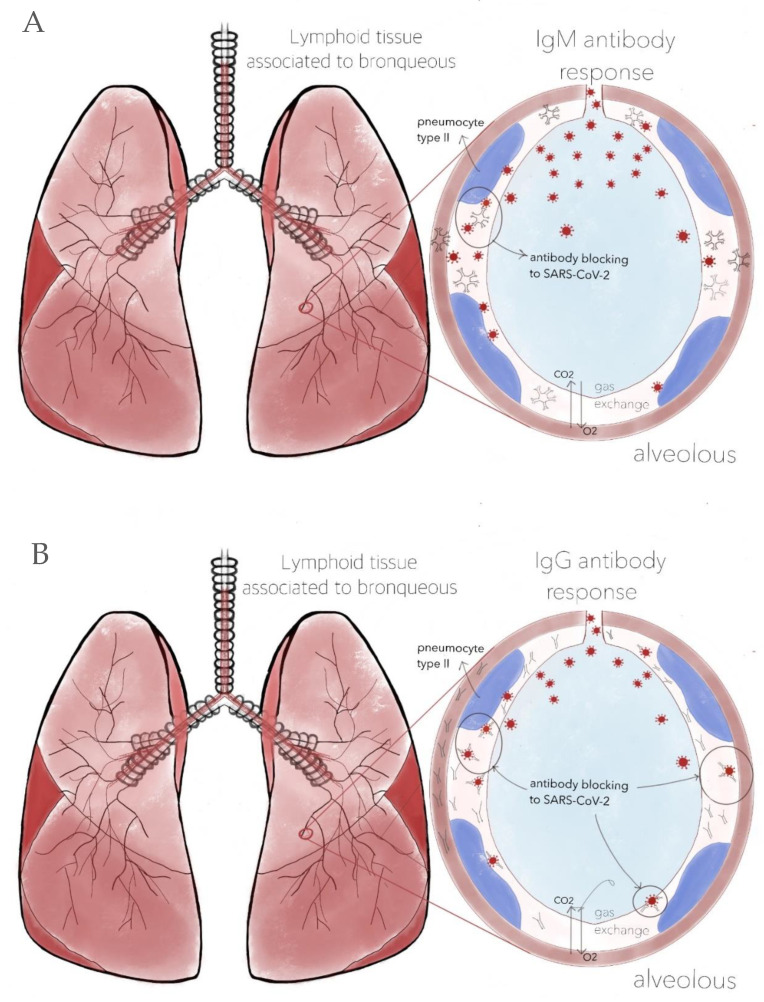
(**A**) Production of IgM antibodies after the first dose. Although there is a considerable quantity of immunoglobulins, their ability to neutralize SARS-CoV-2 is limited due to their size. Additionally, being very heavy, they cannot easily cross the capillary alveolus barrier due to the significant variability in their antigen-binding fragments (Fab). (**B**) IgM and IgG antibodies coexist. The latter are more specific and capable of neutralizing SARS-CoV-2. Due to their lower molecular weight, they manage to cross the capillary alveolus barrier. However, this does not mean that infection cannot occur; it only decreases the risk of developing COVID-19. Therefore, it is necessary to continue with adequate sanitary measures. Design by Carlos U. Torres-Estrella.

**Figure 4 vaccines-10-00414-f004:**
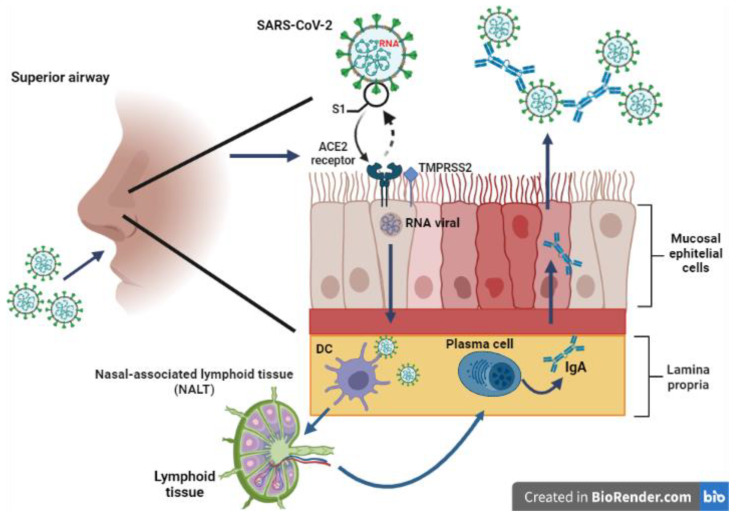
Representation of intranasal immunization, in which a higher production of immunoglobulin IgA is prompted. However, this protein does not remain active for long periods. Design by Thalía F. Camarillo González and Valeria Juárez-García.

**Table 3 vaccines-10-00414-t003:** Primary nucleic acid-based vaccines and viral vectors available for use in some regions of the world.

Type of Vaccine							
	Nucleic Acids		Vector Viral				
Name	Comirnaty (BNT162b2 mRNA)	mRNA-1273	CVnCoV	AZD1222 (ChAdOx1)	Ad5-nCov	Sputnik V (Gam-COVID-Vac)	Ad26.COV2.21S (JNJ-78436735
Manufacturing Company	Pfizer/BioNTech	MODERNA	CureVac/Bayer/GSK/No vartis	AstraZeneca/Oxford	CanSino Biological	Gamaleya Research Institute	Janssen Pharmaceutical Companies of Johnson & Johnson (J & J)
Handling/ Storage	−70 °C up to 6 months, 2–8 °C up to 5 days, reconstituted up to 6 hrs	−20 °C up to 6 months, 2–8 °C up to 30 days	2–8 °C up to 3 months	2–8 °C	2–8 °C	1st vial frozen at −18 °C 2nd vial lyophilized at 2–8 °C	2–8 °C
Doses required	Three doses Second dose 21–42 days after the first dose Third dose 6 to 12 months after the second dose	Second dose 28 days after the first one	Second dose 28 days after the first one	Second dose 28 days after the first one	Single dose	Second dose 21 days after the first one	Single dose *
Immunization per dose	100 µg 30 µg (3rd doses)	30 µg	12 µg	0.5 × 10^11^ Vp	0.5 × 10^11^ Vp	0.5 mL	0.5 × 10^11^ Vp
% Efficacy in preventing infection	95%	94.1%	Phase III data to be published	82.4%	65.28%	92%	72% in the USA 61% in Latin America
Observations	It contains a strand of mRNA that codes for the protein S “Spike” wrapped in a lipid nanoparticle using polyethylene glycol as a stabilizing agent. The third dose is being evaluated in patients 18–55 years and 65–85 years.	It contains a strand of mRNA that codes for the protein S “Spike” wrapped in a lipid nanoparticle.	It contains a strand of mRNA that codes for the protein S “Spike” wrapped in a lipid nanoparticle. Mexico is one of the countries selected for phase III.	Chimpanzee adenovirus containing mRNA encoding protein S “Spike.”	Modified adenovirus serotype Ad5 containing mRNA encoding protein S “Spike.”	The first vial is a modified adenovirus serotype Ad26. The second one is a modified adenovirus serotype Ad5. Both contain double-stranded DNA with the S gene for the “Spike” protein.	Modified adenovirus serotype Ad26 containing double-stranded DNA with the “Spike” protein S gene.

EMA: European Medicines Agency; Vp: Viral particles; mRNA: Ribonucleic acid of the messenger type; DNA: Deoxyribonucleic Acid; * J & J Pharmaceuticals indicates that two doses may be required depending on the patient’s needs and the health care provider’s determination. They all contain the “S” gene in the form of mRNA or DNA [21,22,23,24,25,26,27,28,29,30,31,32].

**Table 4 vaccines-10-00414-t004:** Primary vaccines based on the attenuated SARS-CoV-2 virus and protein “S” subunits available for use in some regions of the world.

Type of Vaccine						
Characteristics	Attenuated Pathogen			Protein Subunities		
Name	CoronaVac	Covaxin (BBV152 A, B, C)	Not Available	BBIBP-CorV	NVX-CoV2373	ZF2001
Manufacturing Company/Institution	Sinovac	Bharat Biotech/Indian Council of Medical Research	Sinopharm/Wuhan Institute of Biological Products	Sinopharm/Beijing Institute of Biological Products	NOVAVAX	Anhui Zhifei Longcom Biopharmaceutical Co./Government of Uzbekistan
Handling/Storage	2–8 °C	2–8 °C	2–8 °C	2–8 °C	2–8 °C	2–8 °C
Doses required	Second dose 14 days after the first one	Second doses 28 days after the first one	Second dose 21 days after the first one	Second dose 21 days after the first one	Second dose 21 days after the first one	2–3 doses 28 days after the first one
Immunization per dose	3 µg	3 µg	Unknown	4 µg	5 µg SARS-CoV-2 rS + 50 µg of Matrix-M1 adjuvant	25 µg/0.5 mL
% Efficacy in preventing infection	83.7% in Turkey 50.3% in Brazil	81%	72.5%	79.34%	96% Original coronavirus 86% variant B.1.1.7 49% variant B.1.351	Not reported
Observations	-	-	-	-	Nanoparticles containing the protein subunit S.	Recombinant origin using CHO cell line to express protein S.

CHO: Chinese Hamster Ovary; Adjuvant: A molecule that helps increase the immune response; B.1.1.7: British variant; B.1.351: South African variant [16,24,33,34,35,36].

**Table 5 vaccines-10-00414-t005:** Data reported on humoral and cellular immunity generated by vaccines administered so far to the population worldwide.

Vaccine	Maximum Antibodies	Type of Immunity Reported	Detection Method
RNm-1273 NIAID Moderna	Antibodies have been reported six months after vaccination	CD4 + T H 1 cells (TNF-α> IL-2> IFN-γ), low expression of TH2 cytokines (IL-4 and IL-13) and detectable CD8 + T cell responses	ELISA
NT162b1 Pfizer/BioNTech	Antibody rise 14 days after the booster dose	Concurrent production of neutralizing antibodies, activation of CD4 + T lymphocytes biased to TH1 with little response of TH2 (IL-4) and CD8+, virus-specific, and the solid release of immunomodulatory cytokines such as IFNγ.	Flow cytometry, IFNγ ELISpot and cytokine profile
CanSino	IgG antibodies at 28 days. Neutralizing antibodies at 8 weeks.	CD4 + and CD8 + T cells produced IFN-γ, TNF-α, and IL-2, with a large proportion of both subsets of T cells being unique IFN-γ producers. Strong IgG1 and IgG2 responses.	ELISA IgG NAb by virus-specific microneutralization
ChAdOx1 CoV-19/AZD1222 AstraZeneca	T-cell response from day 7, peaking on day 14 and remaining detectable until day 56. The last analysis detected IgG being at its peak on day 28 and remaining until day 56.	CD4 T + predominantly secreted Th1 cytokines (IFN-γ, IL-2, and TNF-α) rather than Th2 (IL-5 and IL-13).	Detection by IFN-γ ELISPOT assay before and after vaccination and flow cytometry.
VX-CoV237 (Novavax)	IgG anti-S: 31/32 days after one dose. Neutralizing antibodies: 21–28 days after the first dose. IgG anti-S: Titers increased 1 to 35-fold within ten days after second dose immunization.	Induced CD4 + and CD8 + T cell response. Matrix-M adjuvant improves the development of Tfh cells and GC B. cells (Vaccine in phase III of clinical trials).	ELISA
26.COV2.S Janssen/Johnson & Johnson	The first dose showed neutralizing antibodies on days 57 and 71. The second dose showed an increase in neutralizing antibody titers at day 57	Central memory CD27 +/CD45RA−/CD4 + and CD8 + T cell response. Biased TH1 cellular immune response.	LISA, ELISPOT, and IFN-γ assays for cellular immune response. Intracellular cytokine staining for CD4 + and CD8 + T cells.

ELISPOT: Enzyme-Linked Dot Immunoadsorption Assay; ELISA: Enzyme-Linked Immunosorbent Assay; NAb: Neutralizing antibodies; [21,23,25,26,28,29,30,31,32].

**Table 6 vaccines-10-00414-t006:** Leading companies/institutions developing projects to produce a COVID-19/vaccine in Mexico.

Institution/Company	Financing	Type of Vaccine
Avimex^®^, Universidad Nacional Autónoma de México and Instituo Mexicano del Seguro Social.	AMEXCID, CONACyT and SECTEI	Viral vector with nucleic acids. Veterinary platform.
Instituto de Biotecnología, Universidad Nacional Autónoma de México	AMEXCID, CONACyT and SECTEI	Viral vector
Universidad Autónoma de Querétaro and Instituto Politécnico Nacional	AMEXCID, CONACyT and SECTEI.	Viral vector
Universidad Autónoma de Baja California and Tecnológico de Monterrey	AMEXCID, CONACyT and SECTEI	Synthetic nanoparticle

AMEXCID: Agencia Mexicana de Cooperación Internacional para el Desarrollo, CONACyT: Consejo Nacional de Ciencia y Tecnología, SECTEI: Secretaria de Educación, Ciencia, Tecnología e Innovación.

## Data Availability

Not applicable.

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
