# Peer review of "Vaccines Against COVID-19: A Review"

_vaccines, 2022, doi:10.3390/vaccines10030414_

Round 1
Reviewer 1 Report
Immunogenicity in vaccines against COVID-19.
This review is not oriented really to the immunogenicity of covid vaccines only. It provides a large portfolio of information, nevertheless, this is more superficial and the level of knowledge is in some parts of the manuscript like for medical students. In some parts, the manuscript provides also a historical view on vaccination.
On the other hand, the manuscript brings at least in tables comprehensive information about available vaccines.
This is a very large review, which can be appreciated by medical readers outside of vaccines specialization like surgeons. But for experts in vaccines, this review doesn't bring a high level of information.
There is necessary within the text to do many corrections, like:
Line 287 Please correct the following in the table: Advantajes Disadvantajes
Author Response
Reviewer 1
This review is not oriented really to the immunogenicity of covid vaccines only. It provides a large portfolio of information, nevertheless, this is more superficial and the level of knowledge is in some parts of the manuscript like for medical students. In some parts, the manuscript provides also a historical view on vaccination.
On the other hand, the manuscript brings at least in tables comprehensive information about available vaccines.
This is a very large review, which can be appreciated by medical readers outside of vaccines specialization like surgeons. But for experts in vaccines, this review doesn't bring a high level of information.
Answer: We decided to change the main title attending the major observation for this work. We also made adjustments to improve the clarity of the content and avoid disturbing the principal purpose of providing exhaustive information about COVID-19 vaccines.
There is necessary within the text to do many corrections, like:
Line 287 Please correct the following in the table: Advantajes Disadvantajes
Answer: We have changed the words below line 217 in Table 2.
Reviewer 2 Report
Review of MS# 1584467 for Vaccines.
Immunogenicity in vaccines against COVID-19.
Carlos U. Torres-Estrella, María del Rocío Reyes-Montes, Esperanza Duarte-Escalante, Mónica Sierra Martínez, María Guadalupe Frías-De-León, and Gustavo Acosta-Altamirano
This extremely extensive and wide-ranging review has a number of very laudable aims. It is well written, easy to read and well referenced and up to date. It provides a very good introduction to the basics of the immune responses to pathogens, especially viruses and the various mechanisms employed to generate vaccines. It gives a very good account of the various types of vaccines currently generally used against virus infections. It then highlights the various platforms employed to produce potential vaccines for COVID19. Going on to highlight the situation in the authors location of Mexico. It finished with some well-made points on the need for further vaccine and antiviral development.
My main criticism is that the resulting manuscript is not very focused on the topic of the title, especially in the first half. While being very informative on the immunology of virus infections and generation of vaccines in general, sections 1 – 6 hardly make specific mention to Sars-CoV-2.
I suggest that the authors need to decide if this is a general review of immunology of vaccines to viruses in general – in which case a more general title would be suitable. Or a specific review of “Immunogenicity in vaccines against COVID-19”, in which case sections 1 – 6 need to be rewritten with a much more specific focus on this disease.
I also have the following specific comments on the text:
Line 71: More correctly “Methodology”.
Line 77: “both in English and Spanish” – does this mean references in available in either language or only ones with versions in both languages. I suspect the former is correct, but text is unclear.
Line 81: Calling this section results seems unnecessary as all the sections 3 – 12 fits into this category. I suggest something like – “Immune response to viruses” or such.
Lines 94 – 96: Better explanation needed. Surely endosomes are intracellular.
Line 104: Do you actually mean budding into the host cell or from the host cell?
Line 126 – 129: Provide references for assertions that “it has been observed that their levels decrease in critically ill patients” and “A B-cell decrease has also been observed”.
Figure 1: This is a good illustration of the timeline for vaccine development in general. But for the vaccines for COVID19 a much more accelerated approach has been employed. This should be compared and contrasted in the figure and fully explained in the text to make this relevant to the topic of the manuscript.
Figure 2 and 3 should be combined into one figure panels A and B.
Table 3: Something strange has happened with the formatting of the columns.
Lines 407 – 409: Not that the Pfizer-BioNtech vaccines was granted full FDA licensure in August 23rd 2021.
Section 12: Make it clear what date these values apply to.
Lines 571 – 573: Surely the use of adjuvants will only be applicable to protein-based vaccines not the mRNA and viral vector vaccines. Make this clear.
Author Response
Reviewer 2:
This extremely extensive and wide-ranging review has a number of very laudable aims. It is well written, easy to read and well referenced and up to date. It provides a very good introduction to the basics of the immune responses to pathogens, especially viruses and the various mechanisms employed to generate vaccines. It gives a very good account of the various types of vaccines currently generally used against virus infections. It then highlights the various platforms employed to produce potential vaccines for COVID19. Going on to highlight the situation in the authors location of Mexico. It finished with some well-made points on the need for further vaccine and antiviral development.
My main criticism is that the resulting manuscript is not very focused on the topic of the title, especially in the first half. While being very informative on the immunology of virus infections and generation of vaccines in general, sections 1 – 6 hardly make specific mention to Sars-CoV-2. I suggest that the authors need to decide if this is a general review of immunology of vaccines to viruses in general – in which case a more general title would be suitable. Or a specific review of "Immunogenicity in vaccines against COVID-19", in which case sections 1 – 6 need to be rewritten with a much more specific focus on this disease.
Answer: We decided to change the main title attending the major observation for this work. We also made adjustments to improve the clarity of the content and avoid disturbing the principal purpose of providing exhaustive information about COVID-19 vaccines.
I also have the following specific comments on the text:
Line 71: More correctly "Methodology".
Answer: "Materials and methods" were changed to "Methodology" (Line 70).
Line 77: "both in English and Spanish" – does this mean references in available in either language or only ones with versions in both languages. I suspect the former is correct, but text is unclear.
Answer: We included references available in either language. The latter was clarified in the text (line 75).
Line 81: Calling this section results seems unnecessary as all the sections 3 – 12 fits into this category. I suggest something like – "Immune response to viruses" or such.
Answer: Attending the suggestion, sections 3-5 were rewritten to improve the clarity in every section, starting with "Immune response to SARS-CoV-2" at line 79.
Lines 94 – 96: Better explanation needed. Surely endosomes are intracellular.
Answer: To elude the general immunology review, we deleted the section that tackled this topic.
Line 104: Do you actually mean budding into the host cell or from the host cell?
Answer: To ensure the correct understanding for readers, we changed the term "budding" for "lysis of the host cell" at line 87 as it was unclear.
Line 126 – 129: Provide references for assertions that "it has been observed that their levels decrease in critically ill patients" and "A B-cell decrease has also been observed."
Answer: The statements have been cited accordingly at line 112.
Figure 1: This is a good illustration of the timeline for vaccine development in general. But for the vaccines for COVID19 a much more accelerated approach has been employed. This should be compared and contrasted in the figure and fully explained in the text to make this relevant to the topic of the manuscript.
Answer: Figure 1 was modified to enhance the reader's understanding. Also, we added an explanation between lines 200-206 for a better interpretation.
Figure 2 and 3 should be combined into one figure panels A and B.
Answer: Figures 2 and 3 were joined into one panel 2 a-b, with the appropriate description of every picture in lines 218-225.305-306.
Table 3: Something strange has happened with the formatting of the columns.
Answer: The mistake was fixed between the first and second columns. The formatting of Table 3, line 327, is now correct.
Lines 407 – 409: Not that the Pfizer-BioNtech vaccines was granted full FDA licensure in August 23rd 2021.
Answer: Thank you for your comment. We have updated the vaccines' regulatory status, including the MODERNA vaccine, in lines 338-340.
Section 12: Make it clear what date these values apply to.
Answer: The misunderstanding was explained in lines 448-452.
Lines 571 – 573: Surely the use of adjuvants will only be applicable to protein-based vaccines not the mRNA and viral vector vaccines. Make this clear.
Answer: The suggestion was clarified in lines 511-513.
Reviewer 3 Report
This paper is an interesting combination of very general immunology, some broad vaccinology and some specific information about COVID-19 and SARS-CoV2. The writing is generally good, but the paper is hard to follow because in is not focused and meanders between topics. I would recommend that the authors reduce the scope of the immunology/vaccinology. It is too broad to be useful as a review. I would recommend just sticking to reviewing the immunogenicity and clinical effectiveness of the SARS-CoV2 vaccines. It would be appropriate to then speculate of the nature of the immune responses to the vaccines.
The vaccine immunogenicity review can be a very interesting paper.
Author Response
Reviewer 3:
This paper is an interesting combination of very general immunology, some broad vaccinology and some specific information about COVID-19 and SARS-CoV2. The writing is generally good, but the paper is hard to follow because in is not focused and meanders between topics. I would recommend that the authors reduce the scope of the immunology/vaccinology. It is too broad to be useful as a review. I would recommend just sticking to reviewing the immunogenicity and clinical effectiveness of the SARS-CoV2 vaccines. It would be appropriate to then speculate of the nature of the immune responses to the vaccines. The vaccine immunogenicity review can be a very interesting paper.
Answer: We decided to change the main title attending the major observation for this work. We also made adjustments to improve the clarity of the content and avoid disturbing the principal purpose of providing exhaustive information about COVID-19 vaccines.
Round 2
Reviewer 2 Report
I note that the authors have made a good attempt at addressing many the specific comment made by myself and the other reviewers, I thank them for this.
However, while the latest version of the manuscript is significant improved in several ways, I still feel that my main concern on the lack of specific focus of much of the text directly on COVID19 vaccines, especially the first half has not been sufficiently addressed.
I feel that this manuscript has the potential to be a very good review and resource but requires a major restructuring. For that reason, I am recommending that the current version be rejected, and the authors encouraged to withdraw this version and take time to come back with a much more focused manuscript for resubmission.
Author Response
Reviewer 2:
I note that the authors have made a good attempt at addressing many the specific comment made by myself and the other reviewers, I thank them for this.
However, while the latest version of the manuscript is significant improved in several ways, I still feel that my main concern on the lack of specific focus of much of the text directly on COVID19 vaccines, especially the first half has not been sufficiently addressed.
I feel that this manuscript has the potential to be a very good review and resource but requires a major restructuring. For that reason, I am recommending that the current version be rejected, and the authors encouraged to withdraw this version and take time to come back with a much more focused manuscript for resubmission.
Answer: We have decided to change the manuscript's structure to make the reading clearer and avoid disturbing the principal purpose of providing in-depth information about COVID-19 vaccines.
Reviewer 3 Report
The organization and writing have been improved.
After reading this again, I would suggest the following title: Vaccines against COVID-19 - A Review.
Much of the immunology and vaccinology review is not specific to COVID -19 (the first 6 sections). This could be greatly reduced. The best part of this paper is in sections 7 - 10. But I will leave this to the editors discretion.
The images are of good quality but not specific to COVID-19. The tables have very interesting data and are organized well.
Author Response
Reviewer 3:
The organization and writing have been improved.
After reading this again, I would suggest the following title: Vaccines against COVID-19 - A Review.
Much of the immunology and vaccinology review is not specific to COVID -19 (the first 6 sections). This could be greatly reduced. The best part of this paper is in sections 7 - 10. But I will leave this to the editors discretion.
The images are of good quality but not specific to COVID-19. The tables have very interesting data and are organized well.
Answer: We have decided to change the manuscript's structure to make the reading clearer and avoid disturbing the principal purpose of providing in-depth information about COVID-19 vaccines. Also, we have made some changes in figures 3 and 4 to enhance COVID-19 data.
Round 3
Reviewer 2 Report
I would like to thank the authors for taking on board the suggestions of the reviewers and making a very significant change to the structure and focus of this manuscript. In my opinion it was well worth the efforts. I am also impressed with how quickly the authors manage to make these significant structural changes compared to the last version, this must have been a considerable effort in time.
I think that the new manuscript is a very nice review of COVID-19 vaccines and will be a useful resource for those wishing to get a strong overview of the various vaccines developed.
I ask that the authors make a final careful cheek for typographic errors introduced while making the revisions. Especially the numbering of sections and figures.
When these errors are corrected I think that this manuscript will be suitable for publication.